# The Relationship between Substance Use Stigma and COVID-19 Vaccine Hesitancy

**DOI:** 10.3390/vaccines11071194

**Published:** 2023-07-03

**Authors:** Natasha Powell, Bruce Taylor, Anna Hotton, Phoebe Lamuda, Elizabeth Flanagan, Maria Pyra, Russell Brewer, O’Dell Johnson, Harold A. Pollack, John A. Schneider

**Affiliations:** 1Pritzker School of Medicine, University of Chicago, Chicago, IL 60637, USA; natasha.powell@uchospitals.edu; 2NORC at the University of Chicago, Chicago, IL 60637, USA; taylor-bruce@norc.org (B.T.); lamuda-phoebe@norc.org (P.L.); flanagan-elizabeth@norc.org (E.F.); 3Departments of Medicine and Public Health, University of Chicago, Chicago, IL 60637, USA; ahotton@bsd.uchicago.edu; 4Department of Medical Social Sciences, Northwestern University, Evanston, IL 60201, USA; mariap@howardbrown.org; 5Department of Medicine, University of Chicago, Chicago, IL 60637, USA; rbrewer@bsd.uchicago.edu; 6Southern Public Health and Criminal Justice Research Center, University of Arkansas for Medical Sciences, Little Rock, AR 72205, USA; ojohnson4@uams.edu; 7Crown Family School of Social Work, Policy, and Practice, University of Chicago, Chicago, IL 60637, USA; haroldp@uchicago.edu

**Keywords:** COVID-19, vaccine hesitancy, substance use stigma, opioids, methamphetamine, cocaine

## Abstract

Two parallel public health epidemics affecting the United States include the COVID-19 pandemic and a rise in substance use disorders (SUDs). Limited research has examined the potential relationship between these two epidemics. Our objective was therefore to perform an exploratory study in order to examine the association between public stigma toward people with a past history of opioid, methamphetamine, cocaine, and alcohol use disorder and COVID-19 vaccine hesitancy. A national sample of U.S. adults (N = 6515) completed a survey which assessed the degree of negative perceptions toward individuals with a past history of substance use disorders (referred to as substance use stigma) and COVID-19 vaccine hesitancy, along with variables such as racial prejudice, source of health news, and other demographics. We evaluated four multivariable logistic regression models to predict COVID-19 vaccine hesitancy, utilizing substance use stigma toward opioids, methamphetamine, cocaine, and alcohol use as independent variables. We found that COVID-19 vaccine hesitancy was positively associated with substance use stigma toward opioid (AOR = 1.34, *p* < 0.05), methamphetamine (AOR = 1.40, *p* < 0.01), and cocaine (AOR = 1.28, *p* < 0.05) use, but not alcohol use (AOR = 1.06, n.s.). Predictive models that incorporate substance use stigma may therefore improve our ability to identify individuals that may benefit from vaccine hesitancy interventions. Future research to understand the underlying reasons behind the association between substance use stigma and COVID-19 vaccine hesitancy may help us to construct combined interventions which address belief systems that promote both substance use stigma and vaccine hesitancy.

## 1. Introduction

There are two parallel public health epidemics affecting the United States: the COVID-19 pandemic and a rise in substance use disorders (SUDs). As of September 14, 2022, an estimated 1,059,605 Americans have died from COVID-19 [1], and SUDs remain undertreated even as their prevalence increases in the U.S. population; this accounts for more than 100,000 overdose deaths in the United States annually [2,3]. Additionally, the COVID-19 pandemic itself has been attributed to an increase in substance use [4,5,6]. Vaccination is an essential COVID-19 prevention strategy, just as screening and evidence-based treatment are essential first steps to addressing SUDs [2]. Vaccine hesitancy remains a key obstacle in preventing COVID-19, and similarly, negative perceptions toward individuals with substance use disorders (heretofore referred to as substance use stigma) can limit willingness to allocate and access SUD treatment resources [2]. These factors and beliefs may be interrelated.

To maximize vaccination rates, as well as to minimize substance use stigma, it is important to understand the complex interplay between factors that may influence people’s decisions to refuse the COVID-19 vaccine or to stigmatize substance use. An improved understanding of the behaviors, beliefs, and psychological characteristics of vaccine hesitant and substance use stigmatizing individuals may help to create more structured and targeted interventions to encourage higher rates of vaccination and reduced substance use stigma. It may also be possible to create combined or bundled interventions [7] to simultaneously address both of these important issues. Although studies have examined factors that drive COVID-19 vaccination rates, as well as substance use stigma, there has not yet been an examination of a possible link between these two phenomena. 

Adding this additional factor to our predictive models may also allow us to better understand and address individuals’ specific concerns regarding vaccination. For example, if someone stigmatizes substance use and is vaccine hesitant, it may stem from a distrust of governmental public health measures [8,9]. This exploratory study therefore sought to examine the possible association between stigma toward SUDs—specifically opioid use disorder (OUD), methamphetamine use disorder (MUD), crack cocaine (hereafter referred to as cocaine) use disorder (CUD), and alcohol use disorder (AUD)—and COVID-19 vaccine hesitancy. 

The existing literature provides signs that substance use stigma may be associated with COVID-19 prevention behaviors. Higher levels of stigma toward persons with OUD are associated with greater support for punitive policies, and with reduced support for public health-oriented policies [10]. Lower support for public health policies related to OUDs may also affect perceptions of public health policies and recommendations related to COVID-19, therefore resulting in greater vaccine hesitancy. 

Political conservatism has also been shown to be associated with greater stigma toward opioid addiction [11], and with reduced support for evidence-informed harm reduction and opioid use disorder treatment [12]. Conservatives are more likely to refuse the COVID-19 vaccine [13] and believe that COVID-19 is less severe or the result of a conspiracy [14]. Relatedly, obtaining more news from sources such as Fox News contributes to people feeling less vulnerable to COVID-19 [14]. Low COVID-19 vaccine acceptance rates are correlated with conspiracy beliefs worldwide [15]. Dependence on social media platforms, such as health-related news sources, as opposed to physicians, scientists, and scientific journals is also negatively correlated with planning to get vaccinated against COVID-19 [15,16]. In addition, sources of information about substance use and addiction can influence the degree of substance use stigma. For example, familiarity and a greater degree of contact with substance users—such as working as a mental health and addiction specialist—are associated with fewer negative attitudes toward SUDs [17]. Therefore, firsthand information about substance use may influence the degree of substance use stigma, just as sources of information/news influence COVID-19 protective behaviors. 

Given these observed patterns, we hypothesize that stigma toward OUD, MUD, CUD, and AUD will be positively associated with COVID-19 vaccine hesitancy.

## 2. Materials and Methods

In this exploratory study, we analyzed a cross-sectional sample of 6515 participants drawn from the October–November, 2021 AmeriSpeak^®^ survey, a probability-based non-institutionalized panel of over 45,000 panel members which was designed to be representative of the U.S. household population. The overall weighted household recruitment rate for AmeriSpeak, which includes a second stage of recruitment for initial non-responders to capture harder-to-reach populations, is 37%; this is one of the highest rates for comparable national probability-based household panels [18].

The panel provides sample coverage of approximately 97% of the US household population. Randomly selected households were sampled using area probability and address-based sampling, with known selection probabilities from the National Opinion Research Center (NORC) National Sample Frame. AmeriSpeak then contacted sampled households by U.S. mail, telephone, and face-to-face interviews, thus improving coverage by capturing hard-to-reach respondents.

The cross-sectional survey used in this analysis was offered in English and Spanish. A randomly selected group of AmeriSpeak panelists were emailed with a description of the study, a place to provide their informed consent, and an invitation to complete this survey. Study participants who did not respond to the first invitation were contacted by email and phone multiple times over the study’s 1.5-month intake period. Participants received an incentive worth USD 4 for completing the survey.

This study was carried out in accordance with the approval that the research team received from the NORC at the University of Chicago Institutional Review Board (IRB00000967), under its Federal-wide Assurance (FWA00000142). Voluntary and informed consent was obtained from all participants either via the participant checking a time-stamped box online, for those completing the web version of the survey, or consenting verbally, for those completing a survey by phone or in person.

### 2.1. Measures

#### 2.1.1. Social Stigma toward People Who Use Substances

We developed a 10-item scale adapted from prior stigma survey research [10,19]. The scale was administered separately for different SUDs and we assessed the reliability of each of these scales with a Cronbach’s alpha coefficient. Cronbach’s alpha is a measure of the internal consistency of a scale, expressed as a number between 0 and 1 [20], with scores above 0.7 considered a reliable measure [21]. We found each of our stigma measures had good reliability: opioids (Cronbach’s α = 0.84), methamphetamine (Cronbach’s α = 0.86), cocaine (Cronbach’s α = 0.86), and alcohol (Cronbach’s α = 0.82). For each form of SUD, participants were queried about their willingness to engage with, and perceptions surrounding, individuals with a past history of SUD in the following ways: (1) work with you or (2) marry into your family, (3) their likelihood of stealing something to get drugs, and (4) their likelihood of being a high-risk employee in their workplace. Participants were then queried about their willingness to engage with, and perceptions surrounding, individuals with a current history of SUD in the same ways. Additionally, the (9) perceived dangerousness and (10) trustworthiness of people with an SUD were assessed. Respondents rated their agreement with each statement on a five-point Likert-type scale (1 = strongly disagree, 2 = somewhat disagree, 3 = neither disagree nor agree, 4 = somewhat agree, and 5 = strongly agree). Four items on this scale were reverse coded before computing the mean of all ten items. A higher score reflects greater substance use stigma.

#### 2.1.2. Vaccine Hesitancy

We used respondents’ answers to the survey question “Now that a vaccine against COVID-19 is available, do you plan to get vaccinated, or not?” to code them as “vaccine hesitant” or “not vaccine hesitant”. Respondents who chose “I already got vaccinated” or “Yes, I plan to get a COVID-19 vaccine as soon as possible” were coded as zero, representing “not vaccine hesitant”. Respondents who chose “Yes, I will get a COVID-19 vaccine, but I will wait until I see more proof that it is safe and effective” or “No, I will not get a COVID-19 vaccine” or “Not sure” were coded as one, representing “vaccine hesitant”.

We also included the following covariates in our regression models:

#### 2.1.3. Political Party Affiliation

Current political affiliation was collected as a categorical question on whether the respondent was a Democrat, someone who leaned towards the Democrat party, no declared affiliation/independent, leaning Republican, or Republican.

#### 2.1.4. Color-Blind Racial Attitudes Scale (CoBRAS)

Racially conservative attitudes toward Black Americans were measured using a subscale of the Color-Blind Racial Attitudes Scale (CoBRAS), which has been shown to be associated with higher levels of racial prejudice [22]. The CoBRAS includes eight survey items rated on a five-point scale from “Strongly disagree” to “Strongly agree”, and it included items such as “White people in the U.S. have certain advantages because of the color of their skin,” and “Racial and ethnic minorities do not have the same opportunities as white people in the U.S.” A mean of all items was computed to create the scale, with higher scores representing lower perceived awareness of white racial privilege. The Cronbach’s alpha for this measure was 0.90.

#### 2.1.5. Primary Health News Source 

Respondents selected all options which applied from the following eight categories: Mainstream Media, Government, Social Media, Personal Contacts, Community Sources, Do Not Consume Health-Related News, Left-Leaning Sources, and Right-Leaning Sources.

#### 2.1.6. Blaming People Who Are Addicted to Opioids for Their Problems

Respondents rated their agreement with this statement on a five-point Likert-type scale (1 = strongly disagree to 5 = strongly agree).

#### 2.1.7. Believing People Who Are Addicted to Opioids Are Lazy

Respondents rated their agreement with each statement on a five-point Likert-type scale (as with the above item on blame).

#### 2.1.8. Personal Substance Use

Respondents were asked “have you ever used any of the following substances in your life?” and selected all options which applied from a list of heroin, fentanyl, other opioids, crack or other forms of cocaine, methamphetamines/amphetamines or speed, xanax/benzodiazepines/antianxiety drugs or tranquilizers, or other.

#### 2.1.9. Family/Friend Substance Use

Respondents were asked if family members/friends had used any of the following substances in their lives, and they selected all options which applied from a list of heroin, fentanyl, other opioids, crack or other forms of cocaine, methamphetamines/amphetamines or speed, xanax/benzodiazepines/antianxiety drugs or tranquilizers, or other.

#### 2.1.10. Demographics 

Data were collected on the sociodemographic and background characteristics of respondents, including age (categorized in four categories of 18–29, 30–44, 45–59, and 60+ years old), biological sex at birth, race/ethnicity (categorized into four categories of White, Black, Hispanic, and other), income (categorized into four categories of less than USD 30,000; USD 30,000 to USD 59,999; USD 60,000 to USD 99,999; USD 100,000 or more), education (categorized into four groups of less than high school (HS), HS/GED, College/Associate Degree, and Graduate Degree), religion (categorized into 14 categories of Protestant, Roman Catholic, Mormon, Orthodox, Jewish, Muslim, Buddhist, Hindu, Atheist, Agnostic, Nothing in Particular, Just Christian, Unitarian, and Something Else), and religious service attendance (categorized into never, infrequent/less than once per month, and frequent/at least once per month).

### 2.2. Analysis

Descriptive statistics were computed for our substantive and sociodemographic variables. Statistical analyses were conducted using IBM SPSS 24.1. A significance level of *α* = 0.05 was used, and all statistical tests were two-sided. We estimated four multivariable logistic regression models, using COVID-19 vaccine hesitancy as a dependent variable in each model, and alternating the use of social stigma toward OUD, MUD, CUD, and AUD. In each of these four models, we also included the following covariates which were primarily significant for bivariate analyses and were related to vaccine hesitancy in the literature [23,24]: age, sex, race/ethnicity, political party affiliation, educational attainment, household income, religion, “born-again” or evangelical Christian identity, religious service attendance, primary health news source, blaming people with OUD for their problems, believing people with OUD are lazy, CoBRAS Scale, personal use of substances, family or friend use of substances, and interaction between political party affiliation and substance use stigma.

We then used predicted probability margin plots [25] to demonstrate the presence of interactions between stigma and political party affiliation with regard to vaccine hesitancy.

In all analyses, we applied statistical weights to adjust our data to US census benchmarks, accounting for selection probabilities (balanced by sex, age, education, race/ethnicity, and region) and non-responses (using a response propensity approach calculating the conditional probability that a particular respondent completed the survey given observed covariates) [26,27].

In our analyses we found that respondents and nonrespondents did not significantly differ in terms of most of the demographic variables available for participants and non-participants, such as age, biological sex at birth, and race/ethnicity. We did observe small but statistically significant differences in age and region, as respondents tended to be older and from the Midwest region compared with non-respondents. This difference was adjusted with nonresponse weights and incorporated into the final weights used to weigh the sample data.

Regarding generalizability, we recognize that the literature is extensive with regard to low response rates, and they have potential to bias the estimates from research. Our study addressed the problem of a low response rate by drawing a probability-based nationally representative sample of U.S. households and following industry statistical standards when weighting the data. More specifically, we used statistical weights to adjust our data to US census benchmarks, accounting for selection probabilities (balanced by sex, age, education, race/ethnicity, and region) and non-responses (using a response propensity approach calculating the conditional probability that a particular respondent completed the survey with the observed covariates) [27].

The study sampling weights were derived using a combination of the final panel weight used in all studies (using the AmeriSpeak platform) and the probability of selection (regarding the sampled panel members in our specific study on COVID-19). As not all sampled panel members responded to our survey request, an adjustment was needed to account for, and adjust for, survey non-respondents. This adjustment decreases the potential nonresponse bias associated with sampled panel members who did not complete the survey interview for the study. Thus, the nonresponse-adjusted survey weights for the study were adjusted via a raking ratio method for a general population aged 18 and older. The population totals were associated with the following topline socio-demographic characteristics: age, sex, education, race/Hispanic ethnicity, and Census Division. The population totals were also associated with the following socio-demographic interactions: age × gender, age × race/ethnicity, and race/ethnicity × gender. The weights adjusted to the external population totals were our final study weights. 

At the final stage of weighting, any extreme weights were trimmed based on a criterion of minimizing the mean squared error associated with key survey estimates; then, the weights were re-raked to the same population totals. Raking and re-raking is conducted during the weighting process so that the completed weighted demographic distribution of the survey resembles the demographic distribution in the target population. By aligning the survey respondent demographics with the target population, the key survey items align with the target population.

## 3. Results

All 6515 respondents were included in the final sample, as less than 1% of the data were missing for any survey item. Approximately 40% of the contacted participants from the AmeriSpeak panel completed this project’s survey. We had an overall response rate of about 15% (0.37% panel participation rate * 40% study participation rate). Participants (3449 female and 3057 male) aged 18 and older were included in all analyses. 

Table 1 summarizes the sociodemographic and key characteristics of the respondents. The majority of the respondents identified as white (71.5%). Thirty-eight percent of respondents self-reported as Democrats, 12.3% as leaning Democrat, 15.2% as neither/do not lean/independent, 10.6% as leaning Republican, and 23.9% as Republican. Only 19.7% reported vaccine hesitancy.

Table 2 presents results from regressions on substance use stigma for OUD, CUD, MUD, AUD, and COVID-19 vaccine hesitancy. Controlling for other factors, higher levels of OUD stigma (AOR = 1.34, *p* < 0.05), CUD stigma (AOR = 1.28, *p* < 0.05), and MUD stigma (AOR = 1.40, *p* < 0.01) were associated with greater vaccine hesitancy, whereas higher levels of AUD stigma were not associated with greater vaccine hesitancy (AOR = 1.06, n.s.).

In these regression models, we also examined the effect of the interaction between political party affiliation and substance use stigma on vaccine hesitancy. We examined this because political party affiliation and substance use stigma have been found to be related in the literature [28]. We used predicted probability margin plots [25] (see Figure 1) to demonstrate more clearly the presence of interactions between stigma and political party affiliation regarding vaccine hesitancy. 

Figure 1a shows the predictive margins of vaccine hesitancy with respect to political party affiliation and stigma towards OUD. The *Y* vertical axis represents the probability of vaccine hesitancy (range 0 to 1), and the *X* horizontal axis is the probability of stigma towards opioid use (scale 1 to 5). The confidence bands are calculated pointwise (i.e., without any corrections for multiple testing). 

First, overall vaccine hesitancy is nearly flat (see black line with black circles), with partisan differences, thus yielding stronger and more consistent associations with vaccine hesitance than we observed across the range of expressed substance use stigma. However, the overall effect masks significant interactions with political party affiliation. Second, people with divergent political views mostly have similar values with regard to vaccine hesitancy and are at the highest range of the stigma scale (near 5). Third, although Independents and Republicans seem to have similar levels of vaccine hesitancy with respect to the stigma scale, Democrats (both strong partisan and independent leaning democratic) show lower levels of vaccine hesitancy in the middle range of the scale, with partisan Democrats exhibiting stronger differences than Independents and leaning Democrats. The only group for whom vaccine hesitancy falls below 10% are the partisan Democrats, with values on the opioid use stigma scale falling below 2.5. Fourth, although independents tend to demonstrate reduced vaccine hesitancy with higher values of stigma, the strong partisans, both Republicans and Democrats, demonstrate higher levels of vaccine hesitancy. Fifth, regarding Republicans, their vaccine hesitancy probabilities are fairly flat across all the stigma levels and Republicans are the most vaccine hesitant, irrespective of their stigma level. 

The interactions between political party affiliation and stigma toward AUD, MUD, and CUD—as shown in Figure 1b–d—exhibited the same trends as the interaction between political party affiliation and OUD in Figure 1a, as discussed above. 

## 4. Discussion

In this exploratory cross-sectional study of the association between substance use stigma and vaccine hesitancy, we found that COVID-19 vaccine hesitancy was positively associated with substance use stigma toward opioid, cocaine, and methamphetamine use disorders, but not alcohol use disorder, controlling for the interaction between political party affiliation and stigma.

One possible explanation for our finding that greater AUD stigma was not associated with vaccine hesitancy, but greater OUD, CUD, and MUD stigma, is that AUD may be considered less serious or more familiar than other less commonly used drugs, and therefore, it is less heavily stigmatized overall [29,30]. Internet videos, for example, frequently depict alcohol intoxication as funny or attractive and rarely depict negative clinical outcomes [30]. Opioids, cocaine, and methamphetamine, by contrast, may be more heavily stigmatized in news and social media [31] concurrently with the propagation of vaccine misinformation [32]. 

Our data also suggest that across all four types of SUD stigmas, Republicans were more likely to be vaccine hesitant than Democrats. We then examined the interaction between substance use stigma and political party affiliation when predicting COVID-19 vaccine hesitancy, and we found that for stigma toward OUD, CUD, MUD, and AUD, Republicans tended to be the most vaccine hesitant group, irrespective of their substance use stigma level. Additionally, when Democrats had a low substance use stigma level, they tended to have a low level of vaccine hesitancy compared with other groups, which tended to exhibit a midrange level of vaccine hesitancy, even at low substance use stigma levels. It is also interesting to note that these interactions were fairly similar across all forms of SUD stigma, even stigma toward AUD.

Based on these findings, we believe that that substance use stigma may influence COVID-19 vaccine hesitancy. An improved understanding of this potential link may inform public health efforts to tailor our public health intervention resources to reach and serve groups where they are most needed. For example, by including substance use stigma, along with traditional factors (such as sex, age, race/ethnicity, or political affiliation), in predictive models, it may improve our ability to identify individuals that may benefit from vaccine hesitancy interventions.

Additional research is needed to better understand the underlying reasons why multiple forms of substance use stigma are related to vaccine hesitancy. For example, the association may be partly due to distrust in governmental public health measures, distrust in medicine in general, or fear of foreign substances being introduced into human bodies. Understanding the specific nature of this association may help us both more precisely identify those who are most likely to stigmatize substance use and be vaccine hesitant, and construct combined interventions which can directly address the underlying beliefs that promote stigma and hesitancy. These interventions could be especially important given the increases in COVID-19-related drug and alcohol abuse [6], as there is a possibility that this increase in substance use leads to increased stigma toward substance users, which could serve as a barrier to substance users seeking treatment both for their SUDs and for COVID-19.

Our results must be interpreted in light of several study limitations. Our cross-sectional design limits our ability to draw causal inferences. We also included only four types of substances with which to evaluate “substance use stigma”: opioids, cocaine, methamphetamines, and alcohol. Therefore, we may have failed to capture associations between stigma and vaccine hesitancy for other types of substances (e.g., marijuana). Additionally, our survey results may reflect social desirability or other biases arising from the respondents’ self-reports. The fact that the surveys were self-administered, and almost all web-based, as opposed to over the phone, may have helped to somewhat reduce the effects of social desirability [33,34]; however, this is still a significant limitation of our study. We also failed to measure some potentially important variables to explain alternative hypotheses. The survey items used in this paper were part of a brief survey mechanism used by AmeriSpeak as part of their Omnibus monthly survey. Therefore, the research team was limited in terms of how many questions could be asked of the AmeriSpeak participants. For example, we did not retrospectively measure the number of survey respondents who previously contracted a COVID-19 infection prior to taking the survey. It may be the case that some of the survey respondents previously contracted a COVID-19 infection prior to taking the survey and chose not to be vaccinated because of natural immunity-based reasons, rather than due to vaccine hesitancy. The survey also did not ask about actual vaccination history (e.g., childhood vaccines and age-related vaccines such as shingles). Future studies should include these measures and use them as covariates in the analytic models.

Finally, approximately 40% of the contacted participants from the AmeriSpeak panel completed this project’s survey, thus creating the possibility of non-response bias. We sought to address these potential biases through the use of non-response weights, based on observed socio-demographic factors. 

## 5. Conclusions

In conclusion, our data suggest that COVID-19 vaccine hesitancy was positively associated with substance use stigma toward opioid, methamphetamine, and cocaine use disorders, but not alcohol use disorder. Our exploratory findings underscore the complex interplay of factors related to vaccine hesitancy, partisan affiliation, and substance use stigma, and they highlight that a better understanding of these factors may help us to more effectively promote vaccination and treatment for SUDs throughout the U.S.

## Figures and Tables

**Figure 1 vaccines-11-01194-f001:**
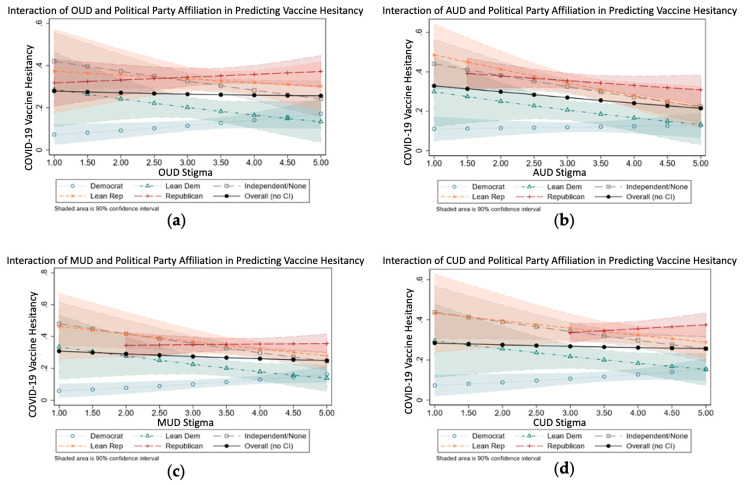
Interactions between substance use stigma and political party affiliation when predicting COVID-19 vaccine hesitancy. These plots were visualized using predictive margins, or adjusted predictions; the predictions were calculated for the predictors of the primary interest (in our case, party and stigma), which varied within their natural ranges, and all other predictors were held at their mean values. Confidence intervals were obtained using the delta method.

**Table 1 vaccines-11-01194-t001:** Characteristics of adult respondents, 19 October–19 November 2021 (*n* = 6515).

Characteristic	Frequency	% of Valid Responses
**Age**		
18–29	1219	18.7%
30–44	1696	26.0%
45–59	1574	24.2%
60+	2027	31.1%
**Sex ^1^**		
Male	3057	47.0%
Female	3449	53.0%
**Race/ethnicity**		
White	4661	71.5%
Hispanic	687	10.5%
Black	697	10.7%
Asian, non-Hispanic	166	2.5%
Other/Two or more	304	4.7%
**Vaccine Hesitancy ^2^**		
Vaccine Hesitant	1277	19.7%
Not Vaccine Hesitant	5211	80.3%
**Religion ^3^**		
Protestant	1718	29.0%
Roman Catholic	1162	19.6%
Mormon	77	1.3%
Orthodox	23	0.4%
Jewish	135	2.3%
Muslim	30	0.5%
Buddhist	57	1.0%
Hindu	29	0.5%
Atheist	299	5.1%
Agnostic	338	5.7%
Nothing in particular	829	14.0%
Just Christian	983	16.6%
Unitarian	49	0.8%
Something else	190	3.2%
**“Born-again” or Evangelical Christian**		
Yes	1007	34.5%
No	1909	65.5%
**Religious Service Attendance**		
Never	2463	37.8%
Less Than Once per Month	2198	33.7%
At Least Once per Month	1825	28.4%
**Political Party Affiliation ^4^**		
Democrat	2466	38.0%
Lean Democrat	796	12.3%
Neither/Do Not Lean/Independent	984	15.2%
Lean Republican	689	10.6%
Republican	1548	23.9%
**Educational Attainment**		
<HS graduate	209	3.2%
HS graduate or equivalent	984	15.1%
Vocational/tech school/some college/associates	2439	37.4%
Bachelor’s Degree	1618	24.8%
Post grad study/professional degree	1265	19.4%
**Household Income**		
<USD 30,000	1363	20.9%
USD 30,000–USD 59,999	1786	27.4%
USD 60,000–USD 99,999	1665	25.6%
USD 100,000+	1701	26.1%
Other	279	4.3%
**Health News Sources ^5^**		
Mainstream Media	4767	73.2%
Government	2767	42.5%
Social Media	2080	31.9%
Personal Contacts	3114	47.8%
Community Sources	882	13.5%
Do Not Consume Health-Related News	582	8.9%
Left-Leaning Sources	2225	34.1%
Right-Leaning Sources	1507	23.1%
**Personal Substance Use in Lifetime ^6^**		
Heroin	209	3.2%
Fentanyl	162	2.5%
Other Opioids	954	14.6%
Crack/Cocaine	643	9.9%
Methamphetamines/Amphetamines/Speed	727	11.2%
Xanax/Benzodiazepenes/Antianxiety Drugs or Tranquilizers	1292	19.8%
Other	279	4.3%
**Family or Friend Substance Use in Lifetime ^7^**		
Heroin	1163	23.2%
Fentanyl	589	12.9%
Other Opioids	1847	39.1%
Crack/Cocaine	1927	36.1%
Methamphetamines/Amphetamines/Speed	1686	33.1%
Xanax/Benzodiazepines/Antianxiety Drugs or Tranquilizers	2420	47.7%
Other	121	3.7%
	**Mean**	**Standard Deviation**
**Blame people with OUD for their problems**	2.96	1.15
**Believe people with OUD are lazy**	2.48	1.04
**CoBRAS Scale ^8^**	2.89	1.04
**OUD Stigma**	3.30	0.66
**MUD Stigma**	3.75	0.70
**CUD Stigma**	3.71	0.71
**AUD Stigma**	3.15	0.64

Abbreviations: OUD, opioid use disorder; MUD, methamphetamine use disorder; CUD, cocaine use disorder; AUD, alcohol use disorder. ^1^ Nine people did not identify as male or female; N = 6506; ^2^ 27 respondents did not answer this question; N = 6488; ^3^ 596 respondents did not answer this question; N = 5919; ^4^ 32 respondents did not answer this question; N = 6483; ^5^ Health News Source category selections were not mutually exclusive; ^6^ Personal Substance Use in Lifetime category selections were not mutually exclusive; ^7^ Family or Friend Substance Use in Lifetime category selections were not mutually exclusive; ^8^ Color-Blind Racial Attitudes Scale with higher scores associated with higher racial prejudice.

**Table 2 vaccines-11-01194-t002:** Multivariable logistic regressions ^1^ for COVID-19 vaccine hesitancy separated by stigma toward type of substance use disorder.

	Vaccine Hesitancy
Type of Stigma	AOR (95% CI)
OUD Stigma (*n* = 5793)	1.34 (1.03, 1.76) *
CUD Stigma (*n* = 5787)	1.28 (1.02, 1.60) *
MUD Stigma (*n* = 5790)	1.40 (1.11, 1.76) **
AUD Stigma (*n* = 5792)	1.06 (0.82, 1.36)

Abbreviations: AOR, adjusted odds ratio; CI, confidence interval; OUD, opioid use disorder; MUD, methamphetamine use disorder; CUD, cocaine use disorder; AUD, alcohol use disorder. * *p* < 0.05; ** *p* < 0.01. ^1^ Full model results can be found in Appendix A (Table A1, Table A2, Table A3 and Table A4).

## Data Availability

Data will be held in the data commons as part of the Justice Community Opioid Innovation Network (JCOIN): https://www.jcoinctc.org/.

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
