# Peer review of "The Relationship between Substance Use Stigma and COVID-19 Vaccine Hesitancy"

_vaccines, 2023, doi:10.3390/vaccines11071194_

Round 1

Reviewer 1 Report

In this cross-sectional survey study, Powell et al. examined the potential association between substance use stigma and Covid-19 vaccine hesitancy using multivariate logistic regression modeling. The authors found that substance use stigma and Covid-19 vaccine hesitancy were positively associated for opioids, methamphetamine, and cocaine use disorders, but not for alcohol. The research question addressed in this study is novel and warrants investigation given the impact of Covid-19 pandemic and substance abuse in the United States. Specific comments are provided below.

(1) The authors use Cronbach's alpha as a measure of instrument reliability, but how were the reliability coefficients for the different substance use disorders generated?

(2) The authors discuss in their introduction how political party affiliation and primary health news source contribute to Covid-19 vaccine hesitancy. Do the authors know how many survey respondents previously had a Covid-19 infection prior to taking the survey and chose not to be vaccinated because of natural immunity rather than due to vaccine hesitancy? The survey instrument does not seem to take this possibility into account.  Why?

Reviewer 2 Report

The manuscript of Powell and her co-authors is well-written. The study is very interesting. Indeed, the news about the developed vaccines in the middle of the pandemic and their “emergency use” status created additional anxiety and fear, followed by alcohol consumption increase and substance use disorder. Unfortunately, this topic was not sufficiently covered and studied by the scientific community. The authors have studied this issue in detail and the conclusions of the manuscript correspond to the aims and funding. Since the list of references is very limited, I would suggest expanding the discussion by adding the previously published related studies. There are some of them: 

- COVID-19 Vaccination and Alcohol Consumption: Justification of Risks (10.3390/pathogens12020163)

- Exploration of Correlations between COVID-19 Vaccination Choice and Public Mental Health Using Google Trend Search (10.3390/vaccines10122173)

- Substance use and abuse, COVID-19-related distress, and disregard for social distancing: A network analysis (10.1016/j.addbeh.2020.106754)

-  COVID-19 vaccine effectiveness in people with substance use disorder (10.1016/S2215-0366(23)00149-9)

Reviewer 3 Report

Important topic. Whether it can be best assessed by low response surveys is the key question you need to answer]

Abstract

Please make it clear that it is the attitudes of the sample and their perceptions surrounding individuals with a past history of SUD and not the actual SUD persons: “perceptions surrounding individuals with a past history of SUD in the following 130 ways:“

Methods

“The overall weighted household recruitment rate for AmeriSpeak, which includes a 102 second stage of recruitment for initial non-responders to capture harder-to-reach popula- 103 tions is 37%, one of the highest for comparable national probability-based household pan- 104 els”.

[low response rates have been a problem for surveys for decades and commented on extensively by sociologists. What is the generalisability of your study?]

[what did you find about the non-respondents when you analysed the non-responses to the requests to participate?]

Results

[Why did you not ask about actual vaccination history (childhood vaccines, influenza, … and age- related vaccines such as shingles? And use as a covariate in your analyses]

“Finally, not all respondents who we 319 sampled and contacted completed the survey, creating the possibility of non-response 320 bias. We sought to address these potential biases through the use of non-response weights, 321 based on observed socio-demographic factors.”

[this is the key weakness of your study. Please state “37% response rate” rather than the misleading “not all.”

[Please explain how the application of non-response weights affected your results. Where did you derive them and what is their validity and reliability?]

“Additionally, 316 our survey results may reflect social desirability or other biases arising from respondent 317 self-report.”

[please explain from the survey literature how “self report affects social desirability”]

[Your conclusions based on a 37% response rate should be much more guarded].

Typo

“was less than 1% 219 data” (Latin: datum = singular, data = plural]

Round 2

Reviewer 3 Report

Thank you for your very detailed responses to the reviewer's suggestions, I agree that dealing with low response rates is very difficult.

I appreciated your very detailed replies. Please go over your Ms. and be sure you included all your valuable comments. They will be helpful to the authors of studies using similar databases. 

Author Response

Response to Reviewer 3 Comments

Point 1: Thank you for your very detailed responses to the reviewer's suggestions, I agree that dealing with low response rates is very difficult. I appreciated your very detailed replies. Please go over your Ms. and be sure you included all your valuable comments. They will be helpful to the authors of studies using similar databases. 

Response 1: Thank you for your kind comments. We have added the remainder of our comments to the text of the manuscript.